# A Black Ice Detection Method Based on 1-Dimensional CNN Using mmWave Sensor Backscattering

**Jaewook Kim** [1,†]**, Eunkyung Kim** [2,†] **and Dongwan Kim** [3,*]

1   Department of ICT Integrated Ocean-Front Smart City Engineering, Dong-A University, 37, Nakdong-Daero 550 beon-gil, Saha-gu, Busan 49315, Korea
2   Department of Artificial Intelligence Software, Hanbat National University, 125, Dongseo-Daero, Yuseong-gu, Daejeon 34158, Korea
3   Department of Electronic Engineering, Dong-A University, 37, Nakdong-Daero 550 beon-gil, Saha-gu, Busan 49315, Korea
*   Correspondence: dongwankim@dau.ac.kr
†   These authors contributed equally to this work.

**Abstract:** Black ice on the road can be dangerous, as it renders the road slippery and is difficult to identify, owing to its transparency. Although studies on black ice detection using cameras, optical sensors, and infrared sensors have been conducted, these sensors have limitations, as they are affected by low light conditions and sunlight. To detect black ice regardless of low light conditions or sunlight, in this study, we incorporate a mmWave sensor that is consistent with varying light conditions. In the proposed method, a frequency modulated continuous wave is transmitted to the surface by the mmWave sensor, and the mmWave sensor backscattering is modulated by the surface medium and roughness. The proposed method also includes preprocessing to calculate the Range-FFT result of the mmWave sensor backscattering and a classification based on a 1-dimensional convolutional neural network to precisely detect the presence of black ice from the Range-FFT result. As a result of the indoor experiment, the proposed black ice detection method achieves an accuracy of 98.2% on dry, wet, and black ice surfaces. Additionally, under low light conditions and in an outdoor environment with sunlight, the proposed method achieves accuracies of 95.6% and 98.5%, respectively.

**Keywords:** black ice; mmWave sensor; FMCW; 1D CNN

## 1. Introduction

Black ice is a thin layer of ice formed on the road surface. When black ice is formed on the road surface, it tends to reduce the friction coefficient of the road. A road with black ice is slippery and prone to safety accidents for pedestrians and automobiles. In addition, due to its transparent nature, black ice is difficult to identify compared to other slippery surfaces, such as wet and snowy roads. According to the Federal Highway Administration, more than 116,800 people are injured in vehicle accidents caused on snowy, slushy, or icy pavements annually in the USA [1].

The conducted studies on black ice detection can be categorized into two types: contact sensor-based methods [2,3] and contactless sensor-based methods [4–9]. In the contact sensor-based methods, to detect black ice, measured temperature and humidity from a sensor installed on the road surface are compared with conditions capable of forming black ice. In contactless sensor-based methods, black ice is detected by utilizing road images obtained by the camera. In addition, black ice detection methods using an IR sensor and optical sensor use the change in water and ice absorption coefficient depending on the wavelength of the incident signal. Table 1 summarizes the black ice detection method using the contact and contactless sensor. However, the sensors used in existing methods have limitations. A camera could be affected by low light conditions [10]. IR sensors and optical sensors also could be affected by sunlight as another resource [11]. Since black ice can form

regardless of light conditions and sunlight, a black ice detection method using a sensor that could generate compelling results in any light conditions is desired.

**Table 1.** Summary of existing sensors used in black ice detection.

| Sensor | Method | Limitation |
|---|---|---|
| Contact sensor [2,3] | Estimate temperature and humidity to verify two parameters meet the black ice forming condition | Hard to manage distributed sensor |
| Camera [4,5] | Detect black ice in vision data of each road surface | Camera is affected by low light conditions |
| IR sensor [6,8], Optical sensor [7,9] | Utilize absorption coefficients that vary with wavelength | IR sensor and Optical sensor are affected by sunlight |

Unlike a camera, IR sensor, and optical sensor, the mmWave sensor is robust under all light conditions [12]. Although the mmWave sensor's data need additional computation to analyze the results, the mmWave sensor can estimate the object's range, velocity, and angle [13], owing to which it has been used in various fields, such as advanced driver assistance systems [14], target detection [15,16], target classification [17], and gesture recognition [18,19]. Therefore, for black ice detection, we utilize the mmWave sensor. Additionally, CNN-based target detection methods using the mmWave sensor have been presented [20,21]. As input data of the CNN model for target detection, the feature map, which is based on two parameters among the target's distance, velocity, and angle values, is utilized. On the other hand, our proposed method utilizes the Range-FFT result as the input data of 1D CNN for black ice detection since the experiment is stationary and the surface is close to the mmWave sensor.

In this paper, we propose a black ice detection method using a mmWave sensor. In the proposed method, mmWave sensor backscattering, which is influenced by the reflected surface medium and roughness, is utilized. After the mmWave sensor transmits the frequency modulated continuous wave (FMCW) toward the surface, the Range-FFT result of the received mmWave sensor backscattering is calculated. To precisely determine the black ice presence based on the Range-FFT result, the 1-dimensional convolutional neural network (1D CNN) model is employed. We evaluate the proposed black ice detection method in experimental environments, including three surfaces (dry, wet, and black ice). The proposed black ice detection method is represented in Figure 1. The primary contributions of this paper are as follows:

1. To our knowledge, our proposed method is the first approach to detecting black ice by using the mmWave sensor. Instead of theoretical analysis, we utilize the black ice detection model based on 1D CNN, which learned the Range-FFT result obtained from the experimental environment.
2. Experiments for evaluating the proposed method are conducted not only in an indoor environment, but also in other environments, where the sensors used may be affected. The experimental results show that the proposed method achieves an accuracy of more than 95%. These experimental results demonstrate the feasibility of black ice detection by using the mmWave sensor.
3. In other black ice detection using a camera [4], they achieved an accuracy of 96.1. Comparing accuracy in the study, it exhibits that the mmWave sensor could detect black ice more precisely.

The remainder of this paper is organized as follows. Background knowledge related to the proposed method is described in Section 2. Then, our proposed black ice detection method is explained in Section 3. In Section 4, the experimental environment for evaluating the proposed black ice detection method is described, and the experimental results are analyzed. Finally, the conclusion is presented in Section 5.

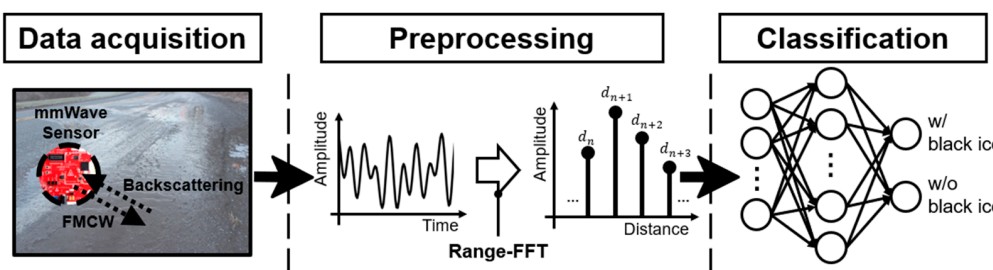

**Figure 1.** The schematic diagram of the proposed black ice detection method.

## 2. Background Knowledge

In the proposed black ice detection method, the mmWave sensor transmits FMCW to the surface. The frequency of FMCW used as the transmitted signal increases linearly with time. After the transmitted signal is backscattered from the surface, the mmWave sensor receives the backscattering. While the signal is backscattered from the surface and received by the mmWave sensor, the frequency of the transmitted signal increases linearly. So, the difference between frequencies of the transmitted signal and the received signal includes frequency components in proportion to distance between the mmWave sensor and the object from which the signal is backscattered. To extract frequency components related to the distance, the transmitted signal, and received signal are used as input data of a frequency mixer. The output of the frequency mixer is the intermediate frequency (IF) signal, and is as follows:

$$x_{IF}(t) = A_{IF} \cos \left\{ 2\pi \left( St_d t + f_c t_d - \frac{S}{2} t_d^2 \right) \right\} \tag{1}$$

where $A_{IF}$ is the amplitude of the IF signal, S is the frequency increase rate, $t_d$ is the time delay that occurs while the signal is transmitted and received, and $f_c$ is the carrier frequency. To analyze the frequency components in the IF signal, a fast Fourier transform (FFT) is applied to the IF signal. An FFT for extracting the distance information from frequency components is called Range-FFT [13]. Since frequency components of the IF signal are in proportion to the distance, the *x*-axis in the FFT may be converted from the frequency domain to distance domain. The range resolution used in the distance domain of the Range-FFT result is expressed as follows:

$$d_{res} = \frac{1}{T_S} \times \frac{1}{N} \times \frac{c}{S} \times \frac{1}{2} \tag{2}$$

where $T_S$ is the duration of the FMCW, c is the speed of light, and N is the number of digitized $x_{IF}(t)$ samples. The FMCW parameters used in this paper are given in Table 2. In the proposed method, black ice presence on the backscattered surface is determined by the Range-FFT result. Meanwhile, the mmWave sensor backscattering can be influenced by the medium and roughness of the surface from which the signal is backscattered [22]. All mediums have an inherent permittivity value, individually. When the signal is incident on the surface, the reflected signal amplitude can be affected by the permittivity of the medium that reflects the transmitted signal [23]. Moreover, when the transmitted signal is reflected, the nature of reflection is decided by the surface roughness. This can be classified into two types as shown in Figure 2. For a smooth surface, the reflected signal components have the same direction. However, if the signal is incident on a rough surface, the reflected signal components scatter in all directions. Considering that the signal amplitude is decided by the amount of signal that enters the receiver, the direction of the reflected signal can be affected by the signal strength [24]. Many other parameters, such as density, surface cover, wetness, etc., also affect signal amplitude [25,26]. By considering all parameters, predicting the black ice presence based on signal amplitude requires extensive computation

power [27]. So, after accumulating the Range-FFT result from experimental surfaces, we train the black ice detection model using obtained data.

**Table 2.** FMCW parameters.

| Parameter | Values |
|---|---|
| Carrier frequency [GHz] | 77 |
| Bandwidth [GHz] | 3.958 |
| Frequency increase rate [MHz/$\mu$s] | 29.982 |
| Duration [$\mu$s] | 132 |
| Sampling frequency [Msps] | 15 |
| ADC sample [EA] | 1536 |

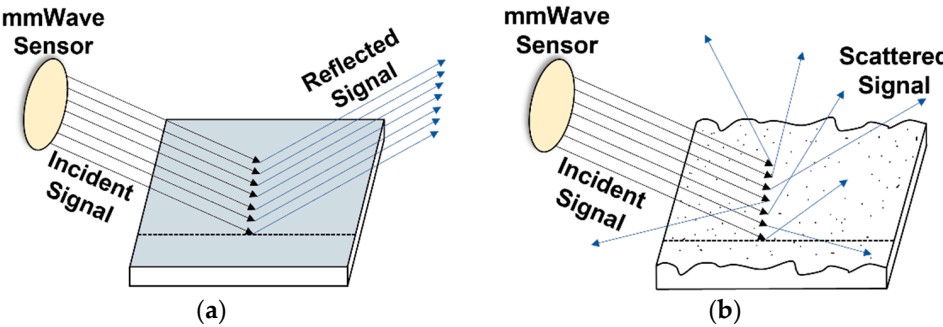

**Figure 2.** The signal reflection for two types of surfaces: (**a**) smooth surface case; (**b**) rough surface case.

### 3. Proposed Black Ice Detection Method

#### 3.1. Data Acquisition

The proposed black ice detection method utilizes the Range-FFT result obtained by the mmWave sensor. As part of the mmWave sensor, we employ Texas Instruments' IWR1443. IWR1443 has three Tx antennas and four Rx antennas. For this study, two Tx antennas and all Rx antennas are used. We also employ a DCA1000EVM to store the sensing results of IWR1443 as bin files on a connected PC using ethernet. In the stored bin file, each sample constituting the digitalized IF signal has a 16-bit complex number form. In this paper, the mmWave sensor is operated by a mmWave studio supported by Texas Instruments. Considering that the maximum bandwidth of the FMCW in IWR443 is 4 GHz, FMCW parameters are set as shown in Table 2 to obtain fine range resolution. When the signal is transmitted by the mmWave sensor using the mmWave studio, the transmitted signal has a constant format. The transmitted signal comprises n frames, and each frame consists of j FMCWs. In this paper, n and j are set as 8 and 128, respectively. After n, j, and the FMCW parameters mentioned in Table 2 are set in mmWave studio, the signal is transmitted toward the surface by the mmWave sensor. Then, the mmWave sensor backscattering from the surface is received.

#### 3.2. Preprocessing

After the transmitted signal is backscattered, the received signal data have a constant format as (the number of used receivers) × (ADC samples × n × j). According to Table 2 and Section 3.1, the data format used in this paper is 4 × 1,572,864. Since the received signal data are handled in the frame unit, the received signal data are divided by frame. Next, the FMCWs that constitute the frame are arranged in the order of sampling data in the time domain. As a representative signal of j FMCWs that constitute the frame, the average signal of j FMCWs is calculated. Next, for the equal utilization of the received signal from all receivers, the average of the received signal from all receivers is calculated. Then, Range-FFT is applied to the representative signal. Concurrently, zero-padding is conducted to make the range resolution finer. Since the range resolution equation is

inversely proportional to the number of ADC samples, zero-padding that puts zero values after the original signal in the time domain can make the range resolution finer. Since the zero-padding factor l is equal to four, the $d_{res}$ is replaced from 4.88 cm to 1.22 cm. The Range-FFT result with zero-padding is as follows:

$$|S| = |S_1, S_2, S_3, \ldots, S_{lN}| \tag{3}$$

where the numbers in the subscript indicate the range bin of the degree of the signal travel distance between the mmWave sensor and the backscattered surface, N represents the original number of ADC samples, and l is the zero-padding factor. Among the Range-FFT results, the partial values that are affected by the surface are extracted. To estimate whether the Range-FFT results are affected by surface or not, the range resolution is utilized. The range resolution is a unit for signal travel distance, not the vertical distance between the mmWave sensor and the surface. So, by measuring the diagonal distance between the mmWave sensor and surface, we can estimate the ideal Range-FFT result affected by surface in the total result. In this manuscript, we extracted more Range-FFT results as a value affected by the surface than the ideal value to ensure that all Range-FFT results affected by the surface are used as input data for the black ice detection model. Before utilizing these obtained data as the input for the black ice detection model, min-max normalization is conducted so that the extracted partial values from the Range-FFT result can be treated without any weighted. The proposed procedure is described in Figure 3. The dataset that is utilized for preprocessing is depicted in Figure 4. Each row of the dataset in Figure 4a,b is used as the input data for the black ice detection method.

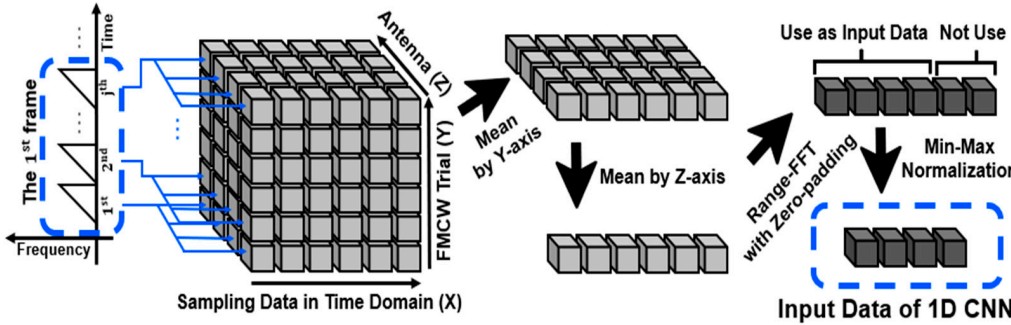

**Figure 3.** The concept diagram of preprocessing procedure.

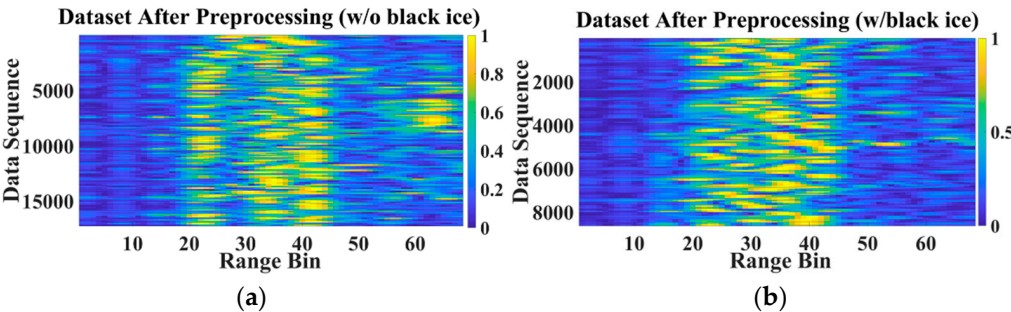

**Figure 4.** The dataset after preprocessing: (**a**) w/o black ice; (**b**) w/black ice.

### 3.3. Classification

In this paper, a black ice detection method based on 1D CNN is utilized to determine the presence of black ice from the surface backscattered signal. 1D CNN is a modified version of 2D CNN that has a dominant performance in image processing [28]. Although 1D CNN has a simple mechanism, it is very effective in analyzing time series data. 1D CNN has two parts, feature extraction and classification, through which it can derive an optimal model, unlike traditional machine learning that requires handcrafted feature extraction [28].

As input data of the 1D CNN model, the partial Range-FFT result obtained from three surfaces (dry, wet, and black ice) in the environment mentioned in Section 4.1 is utilized. The size of the input data is 68 × 1. As input data of the 1D CNN model, the partial Range-FFT result affected by the surface is utilized. The size of the input data is 68 × 1. The feature extraction part includes three convolutional layers, where each convolutional layer is used with batch normalization, rectified linear unit (ReLu) function, and a maxpooling layer. In the convolutional layer, the convolution between input data and each filter is conducted sequentially to extract features from the input data. The size of the filters in all convolutional layers is 3. In each convolutional layer, 15, 20, and 25 filters are used. In the first two convolutional layers, data include zero padding by 1 on both sides. The batch normalization is used to normalize the extracted features using learnable mean and standard deviation. Next, the ReLu function is used as an activation function to reduce the training time for the black ice detection model. The ReLu function removes the negative values while maintaining positive values. Followed by the ReLu function, the maxpooling layer is located to reduce the data size and extract influential features. The size of the filter and its stride value in the first two maxpooling layers are two, and these values in the last maxpooling layer are three.

In the classification part of the model, there is a fully connected layer and softmax function. The fully connected layer makes the decision of black ice presence by adjusting the contribution of the extracted features. In this paper, dropout is applied to the fully connected layer to improve the generalization of the model, and 40% of input data in the fully connected layer is lost intentionally. The softmax function converts the total probability of predicted labels about the input data as 1. As a result, the input data are classified into one of the classes: 'w/black ice' and 'w/o black ice'. Meanwhile, a cross-entropy function is used as a loss function. To minimize the loss function and optimally set many weighted values, an adaptive moment estimation is used as the optimizer. The other parameters used in the model are given in Table 3. The used data rates of training, validation, and testing for the 1D CNN model are 70%, 15%, and 15%, respectively. The 1D CNN model, which is constructed in MATLAB, is represented in Figure 5.

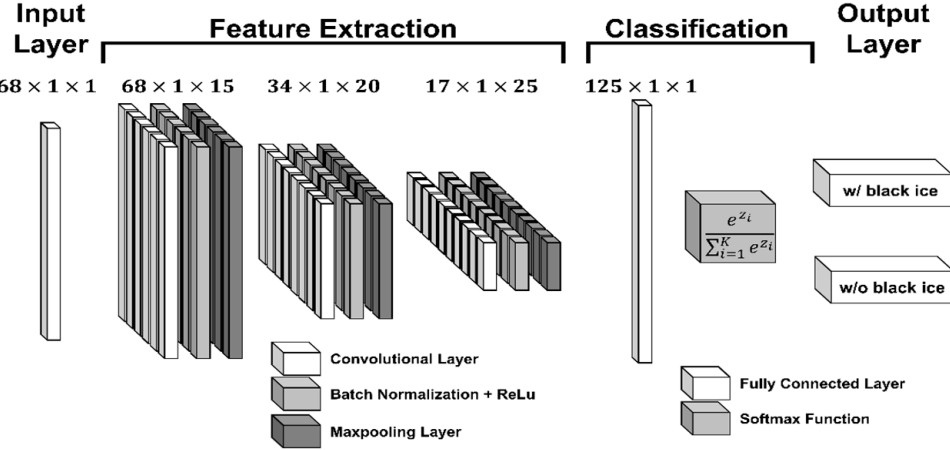

**Figure 5.** The 1D CNN model construction for black ice detection.

**Table 3.** The 1D CNN model parameters.

| Class | Value |
|---|---|
| Learning rate | 0.01 |
| Batch size | 64 |
| Max epoch | 30 |
| Early stopping | 4 |

## 4. Experiment

### 4.1. Experimental Environment

To evaluate the proposed black ice detection method, we set up an indoor experimental environment. A concept diagram for this environment is described in Figure 6. The vertical height of the mmWave sensor from the surface is 75 cm, and the horizontal distance between the mmWave sensor and concrete block is 20 cm. The concrete block is located within the mmWave sensor coverage area.

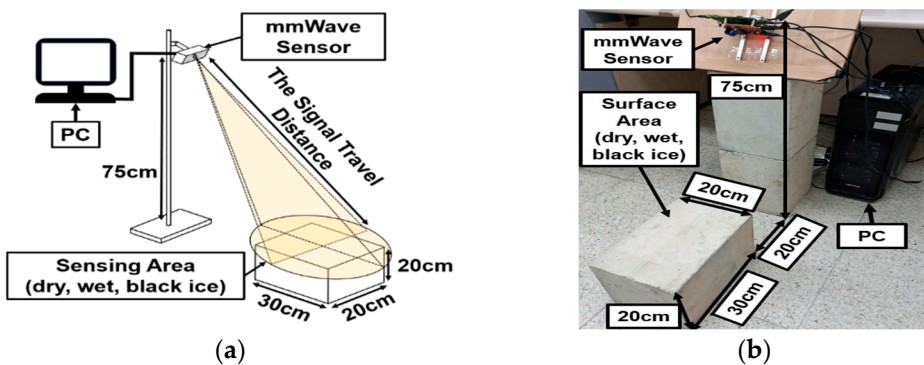

(**a**)  (**b**)

**Figure 6.** The experimental environment: (**a**) concept diagram; (**b**) real environment.

To evaluate the proposed black ice detection method, we set up an indoor experimental environment. A concept diagram for this environment is described in Figure 6. The vertical height of the mmWave sensor from the surface is 75 cm, and the horizontal distance between the mmWave sensor and concrete block is 20 cm. The concrete block is located within the mmWave sensor coverage area. Three types of concrete blocks (dry, wet, and black ice) are used to obtain data from different surfaces. Of the three surfaces, the first two surfaces (dry and wet) are classified as 'w/o black ice'. The third surface is considered as 'w/black ice'. In terms of data accumulation, situations where the data can be biased by a fixed environment should be avoided. So, the concrete block's position is shifted frequently to gather data in variable environments (based on the default position, (1) move to the left 5 cm, (2) move to the right 5 cm, (3) move to the back 5 cm, (4) move to the left 5 cm and back 5 cm, (5) move to the right 5 cm and back 5 cm, (6) move to the back 10 cm, (7) move to the left 5 cm and back 10 cm, (8) move to the right 5 cm and back 10 cm). The concrete black location is adjusted every 20 times the mmWave sensor is operated. In this situation, the data are collected 120 times per position. Considering that eight frames are obtained at once, we can obtain 8640 frames from each surface. After preprocessing is applied to accumulated data, they are utilized for training, validation, and testing of the black ice detection model based on 1D CNN. The black ice thickness is about 1.5 cm. The sandpaper is used to keep the roughness of the black ice surface. Since we have no cooling systems to keep black ice, we employ instant freezing aerosol to maintain the black ice temporarily. As a fundamental measure of preventing the melting of black ice, we had an idle time for every 20 times mmWave sensor operation to freeze the black ice. The indoor temperature of the experimental environment is about 24 °C. Table 4 is a confusion matrix to evaluate our proposed black ice detection method using test data which account for 15% of the dataset obtained in Section 4.1.

**Table 4.** The confusion matrix of experimental results (in the indoor environments).

|  |  | Actual | |
|---|---|---|---|
|  |  | w/Black Ice | w/o Black Ice |
| Predicted | w/black ice | 1211 | 8 |
|  | w/o black ice | 63 | 2606 |

*4.2. Experimental Result*

Accuracy is the rate at which the model correctly predicts the data class for the total data. In the confusion matrix, the row data indicates the prediction result of testing data using our proposed black ice detection method. The column data indicate the actual label of the testing data. Table 4 exhibits that among 3888 test data for evaluating the black ice detection model based on 1D CNN, the model could predict 3817 data. In the experimental result, the proposed black ice detection method achieved an accuracy of 98.2%. In addition, there are two other parameters, precision, and sensitivity, which are used to evaluate the black ice detector performance. Precision is the probability that the data label that was predicted as 'A' ice by the classification model is actually 'A'. Sensitivity is the probability that the classification model predicts 'A' when the actual label of the data is 'A'. In this paper, the black ice detection model achieved precision values of approximately 99.3% and 97.6% in the case of 'w/black ice' and 'w/o black' ice, respectively. Additionally, the obtained sensitivity values in the case of 'w/black ice' and 'w/o black' ice are 95.0% and 99.7%, respectively.

Moreover, experiments are conducted under low light conditions and in an outdoor environment with sunlight. For low light conditions, we use the same experimental environment described in Section 4.1, except for the light conditions. In this experiment, light sources such as windows and LED are blocked. For the outdoor environment with sunlight, again, we construct the outdoor experimental environment with the same conditions in Section 4.1 for full exposure to sunlight. The indoor and outdoor temperatures in the experiment environments are 24 °C and 20 °C, respectively. The total number of mmWave sensor operations in each experiment is 135. The number of mmWave sensor operations to obtain the backscattering from each surface (dry, wet, and black ice) is the same at 45. Since the mmWave sensor transmits 8 frames sequentially, the total data obtained from each experiment are 1080. During data accumulation, the concrete block is moved to avoid accumulating biased data from a fixed environment. After the mmWave sensor backscattering is obtained from both environments, preprocessing is conducted. Then, all preprocessing results are put into the black ice detection model, which is the same as the model drawn in the result in Table 4 to measure the accuracy of the Range-FFT result obtained from the two different environments. Tables 5 and 6 are the confusion matrices obtained for the low light conditions and an outdoor environment with sunlight, respectively. Tables 5 and 6 exhibit that for each 1080 test data obtained from both environments, 1033 and 1064 data are predicted correctly. In the two additional experiments, the proposed black ice detection model achieved accuracies of 95.6% and 98.5%, respectively.

**Table 5.** The confusion matrix of experimental results (in the low-light condition).

|  |  | Actual | |
| --- | --- | --- | --- |
|  |  | w/Black Ice | w/o Black Ice |
| Predicted | w/black ice | 702 | 29 |
|  | w/o black ice | 18 | 331 |

**Table 6.** The confusion matrix of experimental results (in the outdoor environment with sunlight).

|  |  | Actual | |
| --- | --- | --- | --- |
|  |  | w/Black Ice | w/o Black Ice |
| Predicted | w/black ice | 714 | 10 |
|  | w/o black ice | 6 | 350 |

## 5. Conclusions

In this paper, we conducted a study on black ice detection using a mmWave sensor. As input data of the 1D CNN model, we utilized the Range-FFT result calculated from mmWave sensor backscattering that is affected by the surface medium and roughness. We utilized the input data from three types (dry, wet, and black ice) of surfaces in an indoor experimental environment to train the 1D CNN model. The experimental result shows

that our proposed black ice detection method achieves an accuracy of 98.2%. Additionally, this method achieves accuracies of 95.6% and 98.5% under low-light conditions and in the outdoor environment with sunlight, respectively. In our future work, to implement our proposed black ice detection method on a real road, we will accumulate the mmWave sensor backscattering obtained on several kinds of the actual road to train the black ice detection model. Moreover, after designing the ideal black ice model, we will verify the mmWave sensor backscattering obtained from the ideal black ice model by comparing it with the mmWave sensor backscattering from actual experimental environments.

**Author Contributions:** J.K. performed implementation, test, validated the result; E.K. supervised the whole procedure related to this paper; D.K. devised the basic concept of the proposed scheme. All authors have read and agreed to the published version of the manuscript.

**Funding:** This work was supported by the Dong-A University research fund.

**Data Availability Statement:** Not applicable.

**Conflicts of Interest:** The authors declare no conflict of interest.

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
