# Peer review of "A Black Ice Detection Method Based on 1-Dimensional CNN Using mmWave Sensor Backscattering"

_remotesensing, doi:10.3390/rs14205252_

Round 1

Reviewer 1 Report

This paper is trying to present a method based on millimeter wave radar to detect the black ice on the road using 1Dcnn. However, the novelty in this paper is not seen.

The introduction needs to be improved. At least, the current session of introduction should be combined with the section of related works. Also, I believe people have done a lot of research on the target detection and classification using millimeter wave radar. The merits and limits of these methods should be discussed. When talking about the use of cnn, previous work should be reviewed on the application of cnn to road target detection. Also, the authors spend a lot of pages on the literature review. The majority of the work should be focused on their own work and the contribution.

The section of background knowledge is not necessary or at least not to be that long. They are available on many textbooks and papers talking about radar systems. The author should have more effort on how the data is collected, how the system is designed, and how the experiment prepared and performed. 

The processing and classification parts need to be discussed in more details. The authors should point out what is the major difference in their method compared to other methods.

For the experimental results, there is no comparison to other methods. If other methods using millimeter wave radar could not detect the black ice, what is the reason? How does this work overcome the difficulties? These are not reviewed nor discussed in the paper.  Beyond that, how does this method perform on a real road? If experiment is not performed, it is necessary to explain why or discuss about the future plan for real environment test.

Author Response

We really appreciate you for your detailed and thoughtful comments.

We revised the paper according to reviewers comments and summarize modification points in the attached file.

Reviewer 2 Report

Dear authors,

The work is of high quality, especially in the way concepts are explained. The topic is interesting and useful.

The comments are as follows:

Comment 1:

Line 92- You should put period (.) instead semicolon(;).

Comment 2:

Line 108- You should mention and describe Table 1 in the text.

Comment 3:

Line 194- Figure 3 should be more visible and of better quality.

Comment 4:

Line 195- Figure 4 should be more visible and of better quality.

Comment 5:

Line 205 –What means ReLu?  You explained it later, but it needs to be explained the first time it appears in the text.

Comment 6:

Line 244- The space between the number 5 and the centimeter mark is missing.

Comment 7:

It would be desirable if you explained in more detail how you obtained the accuracy of the methods (95,6 and 98,5).

Author Response

(The authors gave the same response as above.)

Reviewer 3 Report

The manuscript titled 'A black ice detection method based on 1-dimensional CNN using mmWave sensor backscattering' introduces a method for black ice detection using mmWave sensors with a CNN algorithm. The authors designed an experimental setup and did many measurments to draw their conclusion. However, I think the manuscript needs major revisions before reconsideration for acceptance or not. 

1. The machanism behind this detection method by mmWave sensors is required, which is not mentioned in this manuscript.

2. More details on the experimental environment are required, such as the operating frequency of the sensor, the temperature, the thickness of the ice layer etc.

3. I cannot understand how the accuracy values in Section 5.2 obtained.

4. Analysis on the confusion matrices in Tables 4, 5, 6 are not enough to demonstrate the efficiency of this CNN method. And how the confusion matrices are conducted is not mentioned here.  

Author Response

(The authors gave the same response as above.)

Round 2

Reviewer 1 Report

Authors have explained their ideas much better than the previous version. I still have a question regarding the manuscript.

In the experiment section, the authors mentioned about performing the experiment with in door and out door conditions. Also the low light and sun light condition.

My question is, if the experiment set up is the same for indoor and out door, how would the experiment be different?

The second question is about light condition. To my knowledge, microwave sensors are not sensitive to light frequency EM waves. Why would it be important to show the different light condition?

My concern is that the reason you obtain a high recognition is due to looking at a given samll sample.

Author Response

Thanks for the reviewer's valuable comments.

We have answered the reviewer's questions in the attachments.

Reviewer 3 Report

The millimeter wave would experience reflection, transmission and volume scattering when it is launched onto the black ice layer. There would also be signal reflection from the interface between the concrete block and the ice layer. Demonstrations on such a physical mechanism as my first comment should be important for a scientific publication instead of a technical report. 

Author Response

(The authors gave the same response as above.)
